# Epidemiologic Factors Supporting Triage of Infected Dog Patients Admitted to a Veterinary Hospital Biological Isolation and Containment Unit

**DOI:** 10.3390/vetsci10030186

**Published:** 2023-03-01

**Authors:** Inês Cunha Machado, Telmo Nunes, Miguel Maximino, João Malato, Luís Tavares, Virgilio Almeida, Nuno Sepúlveda, Solange Gil

**Affiliations:** 1Teaching Hospital, Faculty of Veterinary Medicine, University of Lisbon, Av. Universidade Técnica, 1300-477 Lisboa, Portugal; 2CIISA—Centre for Interdisciplinary Research in Animal Health, Faculty of Veterinary Medicine, University of Lisbon, Av. Universidade Técnica, 1300-477 Lisboa, Portugal; 3Associate Laboratory for Animal and Veterinary Sciences (AL4AnimalS), Av. Universidade Técnica, 1300-477 Lisboa, Portugal; 4Faculty of Veterinary Medicine, University of Lisbon, Av. Universidade Técnica, 1300-477 Lisbon, Portugal; 5Instituto de Medicina Molecular (IMM), Faculdade de Medicina, Universidade de Lisboa, Av. Prof. Egas Moniz, 1649-028 Lisboa, Portugal; 6Centro de Estatística e Aplicações da Universidade de Lisboa, Campo Grande, 1749-016 Lisboa, Portugal; 7Faculty of Mathematics and Information Science, Warsaw University of Technology, Koszykowa 75, 00-662 Warsaw, Poland

**Keywords:** isolation unit, infectious diseases, dogs, triage, hospitalization

## Abstract

**Simple Summary:**

An epidemiologic control of infectious diseases through the creation of infectious disease surveillance systems can be achieved with routine collection and analysis of clinical data. The Biological Isolation and Containment Unit of the teaching hospital from the Faculty of Veterinary Medicine, University of Lisbon is a multispecies facility for the hospitalization of pets with confirmed or suspected infectious diseases and has a database for the routine collection of these patients’ data. With this study, we intend to contribute to optimizing the design of infectious disease control programs and support early triage of these patients through the identification and characterization of the main dog infectious diseases registered over a 7-year period and the identification of potential risk factors for those conditions. The most frequent diagnoses were parvovirosis, leptospirosis, multidrug-resistant (MDR) bacterial infections, and distemper. Some potential risk factors have been identified, with emphasis on age < 2 years old (*p* < 0.001), incomplete vaccination for parvovirosis (*p* < 0.001), age ≥ 10 years old (*p* < 0.001), and presence of a concomitant disorder for MDR-infected cases (*p* = 0.03). Our results, by improving the knowledge about epidemiology and clinical presentation of these diseases, show the value of the collection, analysis, and sharing of clinical data and contribute for the creation of infectious cases triage tools as algorithms.

**Abstract:**

The teaching hospital of the Faculty of Veterinary Medicine at the University of Lisbon hosts a Biological Isolation and Containment Unit (BICU) for the hospitalization of both confirmed and suspected animals of an infectious disease. This study targets the BICU dog population to identify and characterize the most frequent infectious diseases recorded in a 7-year period. Several epidemiologic factors were analyzed for their significance to triage infected cases. During the study period, 534 dogs were admitted, of which 263 (49.3%) had a confirmed infectious disease diagnosis: parvovirosis (49.4%; *n* = 130); leptospirosis (21.7%; *n* = 57); multidrug-resistant (MDR) bacterial infection; (10.6%; *n* = 28), and canine distemper (9.9%; *n* = 26). Several potential risk factors for these diseases were identified: age under 2 years old (*p* < 0.001), incomplete vaccination for parvovirosis (*p* < 0.001), age ≥ 10 years old (*p* < 0.001), and the presence of concomitant disorders for MDR-infected cases (*p* = 0.03). Logistic regression models were constructed to classify cases and controls. The sensitivity and specificity estimates were very high (>0.83) for parvovirosis, MDR, and distemper infections. A lower sensitivity (0.77) was obtained for identifying cases with leptospirosis. In conclusion, infectious diseases are frequent, hence, it is essential to decrease their occurrence through effective preventive measures such as vaccination. The constructed logistic models can also help in triaging admitted dogs with a potential infectious disease.

## 1. Introduction

Veterinary hospitals are transmission hotspots for the spread of different infectious diseases (ID). The interaction between different animal species and their microorganisms is constant over time. For the minimization and elimination of the hazards, measures for infection control and standard operating procedures (SOPs) are essential [1,2,3]. They include administrative measures, cleaning and disinfection protocols, physical barriers, and reference guides for the use of personal protective equipment (PPE) [2,4]. Identification and segregation of both infected and infectious patients is mandatory in SOPs and require the availability of isolation units (IUs) to confine high-risk infectious patients safely, which protects animals, pet owners, hospital staff, and other visitors [5]. Evidence-based health policies are essential to improve healthcare and ID surveillance systems are crucial to establish well-adapted, up-to-date, and rapid response plans for infection control and containment [1,6]. This can be achieved by the routine collection of clinical data from ID patients [2]. The analysis of these data has the potential to generate evidence that could support veterinary practitioners in the differential diagnosis, early detection of a given infectious disease, or even in the prevention of outbreaks [7].

Some of the most reported IDs in dogs that require hospitalization and isolation reported in a wide variety of locations around the world are parvovirosis, leptospirosis, distemper, and infections by multidrug-resistant bacteria (MDR) [4,5]. Parvovirosis is a highly contagious severe gastroenteritis caused by the canine parvovirus type 2 (CPV-2) [8,9]. Leptospirosis is a global waterborne zoonosis caused by spirochetes of the genus *Leptospira* [10]. MDR associated with zoonotic infections is a multifactorial public health concern under the umbrella of the One Health framework. MDR is associated with a large number of species that are resistant to at least one antimicrobial agent in three or more antimicrobial categories [11]. Canine distemper is a serious multisystemic disease caused by a morbillivirus [12].

Circulation of infectious patients must be controlled based on a continuous risk assessment [13]. In the companion animals field, there is a good group of published guidelines concerning hospital infection control focused on disinfection protocols, and the isolation of patients in different kinds of facilities from teaching hospitals to private small practices, covering consultation rooms, surgery sites, and isolation wards applicable worldwide [1,2,13] but there is a lack of evidence-based results in these guidelines, containing real patient data that have identified patterns for the early detection and segregation of potentially infectious patients [13].

This study intends to contribute with real data for a more accurate triage of canine infectious patients, with particular focus on parvovirosis, leptospirosis, MDR infection, and canine distemper cases, in a hospital environment, which can help to fill this gap, and thus reduce hospital-acquired infections. Moreover, as in other areas where the early detection of cases such as that of trauma patients takes place, the development of an early and effective infectious patient management tool can also reduce delays in treatment and lead to subsequent success [14,15,16]. To achieve that, the specific aims of this work were to characterize the infectious dog population admitted to a university-based Biological Isolation and Containment Unit (BICU) over a 7-year period and to identify various determinant factors for the early detection and hospitalization of dogs with ID.

## 2. Materials and Methods

### 2.1. Biological Isolation and Containment Unit under Study

The BICU has been functioning since October 2013 in a building physically separated from the main teaching hospital (TH), from the Faculty of Veterinary Medicine at the University of Lisbon, it is a multispecies facility for the hospitalization of animals that are either confirmed or suspected cases awaiting an ID diagnosis. It has two hospitalization wards for dogs and another two for cats, with capacity for four patients per room. It operates under negative pressure, high-efficiency particulate air (HEPA) filters, a video surveillance system, PPE, and SOP. The BICU receives dogs with gastrointestinal, respiratory, or skin disorders with suspected ID, such as those caused by parvovirus, enteric coronavirus, rotavirus, *Campylobacter* spp., canine distemper, canine infectious tracheobronchitis, canine infectious hepatitis, leptospirosis, dermatitis caused by MDR, dermatophytosis, scabies, and clinical suspicion of involvement of MDR infections, among others. No routine screening tests are performed.

### 2.2. Settings

Data from all dogs admitted to BICU from 1 October 2013 to 30 September 2020 were retrieved from patient medical records stored in the TH management software Guruvet^®^. These records were then compiled and validated into Microsoft^®^ Office Excel 365 spreadsheets.

### 2.3. Participants

Firstly, all the admitted dogs (*n* = 534) were categorized as confirmed ID (*n* = 263), suspected ID (*n* = 182), or non-ID (*n* = 89). The confirmed ID group comprises dogs hospitalized with a definitive ID diagnosis. The suspected ID group encompasses dogs that remained suspect due to an inconclusive diagnostic test. The non-ID group included dogs with a suspicion of ID but with a negative diagnostic test result. Patients are admitted to the BICU with a clinical suspicion of infectious disease based on a set of clinical signs and general laboratory analysis. The suspected ID group includes animals with an inconclusive diagnostic test (e.g., positive antigen test but extremely recent vaccine, positive antibodies test but recent vaccine), animals without diagnostic tests due to owners’ refusal, or that died before sample collection. The confirmed ID group includes patients with typical disease clinical signs and laboratory tests that confirms the suspicion. As for the main diagnosis: in parvovirosis, the presence of gastrointestinal signs such as vomiting and/or diarrhea, and/or leucopenia or neutropenia, anorexia, inappetence; for leptospirosis, renal and/or hepatic and/or respiratory signs; for MDR, signs of bacterial infection namely in the skin and/or urinary tract and in distemper respiratory and gastrointestinal or neurologic or dermatologic signs.

In the case of readmission with the same ID condition, the date of the first hospitalization was used in the data analysis, except for the variable named final outcome when readmissions to the TH are considered.

Data concerning this studied population is presented in Appendix A.

### 2.4. Variables and Measurement

The inclusion criteria for cases of the second phase of the study were a laboratory definitive infectious diagnosis. The confirmed ID group was subject to a descriptive data analysis concerning age, sex, breed, neuter and vaccination status, concomitant diseases, admission date, hospitalization length, and outcome. The same variables were then analyzed separately only for the four most frequent diagnoses: parvovirosis (*n* = 130), leptospirosis (*n* = 57), MDR infection (*n* = 28), and canine distemper (*n* = 26). For the purposes of comparison with the goal of triaging, a twice bigger control group for each of the above infectious diseases was randomly selected from the BICU population during the same period, including animals who did not have the disease, either tested negative or presented with a clinical condition that excluded it. Note that no matching was made between cases and controls in terms of age or gender but the effects of these variables were controlled for in the analysis (see subsection concerning the Statistical Methods).

Data of age were categorized considering Canine Life Stage Guidelines [11]: young (<2 years old), adult (≥2 and <10 years old), and senior (≥10 years old). Given that only a few individuals shared the same breed, dogs were divided into breed and mixed breed.

To achieve a definitive diagnosis, a variety of tests were performed on each dog, according to age, clinical signs, and vaccination status, as shown in Table 1.

The vaccination status was established according to the 2016 Guidelines for the Vaccination of Dogs and Cats from the World Small Animal Veterinary Association (WSAVA) [17], considering the core vaccines plus leptospirosis vaccine due to the risk of exposure within the TH geographic area of intervention [18]. Vaccination was considered up to date if the initial puppy vaccination was completed and there were no missing doses in the 12-month booster or adult revaccinations and out of date if the puppy’s initial vaccination was lacking due to the young age of the puppy or any missing doses in puppy initial vaccination, 12-month booster, or adult revaccinations.

The seasonality, based on the admission date, was included as a categorical covariate concerning the season of the infection irrespective of the year. This covariate had the following categories: cold season, from November to April and warm season from May to October, based on Portuguese climate normalities [19].

### 2.5. Bias

Considering the risk of selection bias in a case–control study from a hospital admitted population, in this study cases and controls were selected considering similar risk of exposure to the main infectious agents in the environment.

### 2.6. Statistical Methods

The statistical analysis was conducted in the R software version 3.6.1 [20]. The results were presented as frequencies of occurrence, percentages for categorical variables, and median with range for continuous variables.

To create triage models for each infectious disease, we adopted a case–control study design where data referring to infected cases were compared to the data of a control group from the BICU, as described in 2.2. This allowed us to construct logistic regression models based on independent Bernoulli trials in order to include the effect of different covariates related to key health determinants. These models enabled us to estimate the probability of a dog entering BICU being a true case of the infectious diseases under analysis as a function of the significant covariates.

All covariates with *p*-values < 0.2 in the variable selection step were included in the final models for each disease [21]. In these final models, *p*-values < 0.05 were indicative of statistical significance of the respective covariate. Coefficient estimate and the respective standard errors were calculated according to the logistic regression models. Tests of multicollinearity between covariates were performed as a model diagnostic step. To assess the predictive performance of the final models, the C-statistic and the area under the receiver operating characteristic curve (AUC) were calculated using the package pROC [22]. Finally, sensitivity and specificity were estimated using the package Optimal Cutpoints [23]. For a given ID, the respective estimates were determining by the point in the ROC curve that maximized the Youden Index.

## 3. Results

### 3.1. Analysis of Dogs with a Confirmed Diagnosis of an Infectious Disease

The initial database comprised a total number of 534 dogs of which 263 (49.3%) had a definitive infectious diagnosis. This infected population was characterized according to the main host and environmental patterns, hospitalization course and the respective clinical outcome, and then separated by diagnosis (Table 2).

At first, a general characterization of dogs with a confirmed infectious diagnosis was performed. The median age was 10 months (0.8 years), with young dogs (<2 years old) being the most represented group. Males, intact, and breed dogs were the most frequent classes. The majority of infected dogs had no updated vaccine status and no concomitant disorders. According to the month of hospitalization (Figure 1), October was the month with the most hospital admissions (14.1%; *n* = 37) and August the month with the fewest hospital admissions (3.4%; *n* = 9) of infected cases. The percentage of discharged dogs was 71.9% (*n* = 189), similar when considering readmissions.

The most frequently diagnosed IDs were parvovirosis (49.4%; *n* = 130), leptospirosis (21.7%; *n* = 57), MDR infection (10.6%; *n* = 28), and canine distemper (9.9%; *n* = 26). The remaining IDs (8.4%: *n* = 22) included dermatophytosis, adenovirus, infectious tracheobronchitis, sarcoptic scabies, herpesvirus, and *Cryptosporidium* spp. The most frequent diagnoses were characterized individually, and potential factors were investigated for an association with these diagnosed infectious diseases (Table 2, Table 3 and Table A1; Figure 1).

### 3.2. Analysis of Dogs with a Confirmed Parvovirosis Diagnosis

For parvovirosis (*n* = 130), the median age was 4 months (0.3 years), young dogs (<2 years old) being the most frequent and significatively associated with an increased risk in the logistic regression model. Males, intact, and breed dogs were more frequent but without statistical significance. A not up-to-date vaccine status was significatively associated with an increased risk in the logistic regression model. Most cases had no concomitant disorder, and this variable was borderline significant in the final logistic regression model. October (16.2%; *n* = 21) and September (15.4%; *n* = 20) recorded most parvovirosis patients’ admissions but all months registered at least one case of this infection (Figure 1) and this variable was not significant in the final model.

The final logistic regression model had an estimated AUC of 0.96 (Figure 2). The sensitivity and specificity estimates were 0.94 and 0.92, respectively, which suggested that this model predicts both infected individuals and controls well. Most patients survived the hospitalization (83.8%).

### 3.3. Analysis of Dogs with a Confirmed Leptospirosis Diagnosis

For leptospirosis (*n* = 57), the median age was 6.0 years (≥2 and <10 years old), adult dogs being the most represented and significantly associated with an increased risk in the logistic regression model. Males, intact, and breed dogs were more frequent. although not significantly associated with this infectious disease. Most of the dogs had no up-to-date vaccination status but it was also not significantly associated with disease. Most infected dogs had no concomitant disorders, and its presence was identified as potentially reducing the risk of leptospirosis in the logistic regression model. March (14.0%; *n* = 21) recorded most patients’ admissions while August did not register any case (Figure 1), but was not significantly associated in the multivariate model.

The final logistic regression model provided a good prediction of dogs with a confirmed Leptospirosis diagnosis when compared to the control group (AUC = 0.79; Figure 2). The sensitivity and specificity were estimated at 0.77 and 0.68, respectively, which meant that this model has a poor prediction of cases. More than half of the cases (54.4%) were discharged from hospitalization with a final survival rate of 50.9%.

### 3.4. Analysis of Dogs with a Confirmed MDR Infection Diagnosis

The median age of dogs with MDR infections (*n* = 28) was 9.5 years old, senior dogs (≥10 years old) being the most represented age group, significatively associated with an increased risk. Males, intact, and breed dogs were more frequent but not statistically significant. Not up-to-date vaccination status was identified as potentially reducing the risk of MDR infection. A concomitant disorder was present in most of the cases and was significatively associated with an increased risk. Most admissions of MDR-infected patients were recorded in March (17.9%; *n* = 5; Figure 1); however, there was at least one case registered in the remaining months, and it was not statistically significant. The final logistic regression model had a very high predictive performance with an AUC of 0.93 (Figure 2). As in the case of parvovirosis, the estimated sensitivity and specificity were very high, 0.83 and 0.95, respectively. Most of the dogs survived the first hospitalization (71.4%), decreasing to 53.6% when considering readmissions. Data concerning types of infection and bacterial organisms are presented in Table A2.

### 3.5. Analysis of Dogs with a Confirmed Distemper Diagnosis

For distemper (*n* = 26), the median age was 1 year, the young being the most represented age group (<2 years old) significatively associated with an increased risk. Males, intact, and mixed breed dogs were more frequent although sex and breed were not significative but being neutered was significatively identified as potentially reducing the risk of distemper. Most of the dogs did not have an updated vaccination status, which was significatively associated with an increased risk. Mainly patients had no concomitant disorders, although not significative in the final model. Most distemper admissions were recorded in January (23.1%; *n* = 6) and no cases were registered in August or September, but without statistical significance (Figure 1; Table A1). The final logistic regression model provided good predictors of both cases and controls (AUC = 0.93; sensitivity = 0.85; specificity = 0.96; Figure 2). Half of the dogs admitted survived the first hospitalization but only 34.6% survived considering readmissions.

## 4. Discussion

This study aimed to characterize the general infectious population admitted at a veterinary teaching hospital-based BICU and to identify disease determinant factors associated with host and environment for early detection and hospitalization of dogs with IDs. The most frequent IDs were parvovirosis and leptospirosis. Parvovirosis is recognized as the most common ID in dogs, widely disseminated in Portugal [8,24,25]. Leptospirosis is one of the most worldwide spread zoonosis, including in Portugal [18,25]. In recent years, a global increasing public health concern related with MDR infections has been documented [2,26]. Distemper being the fourth most frequent ID in the study population was an unexpected result strongly influenced by an outbreak in Lisbon’s Metropolitan Region from 2014 to 2018 [27,28,29].

An overview of the general infectious population identified a young population, probably due to the combined effect of parvovirosis and distemper cases, both associated with young age [8,12,30,31]. Breed dogs were predominant which may reflect breed susceptibility to infectious diseases as suggested by Kim et al. (2018) or might be explained by better socioeconomic conditions of breed dog owners to pay for diagnosis, hospitalization, and treatment [32,33].

The proportion of properly vaccinated dogs in this study was less than 20%, far below the 30–50% estimates for developed countries [17]. This below-average rate may be due to two main reasons: (i) a high frequency of puppies that did not complete the vaccination schedule because they were not yet old enough; (ii) the study period overlaps with the post Eurozone sovereign debt crisis and severe economic constraints may have reduced owners’ compliance in pet preventive care, leading to a decline in vaccination coverage [17].

The monthly ID distribution showed a peak in October, probably due to coexistence of parvovirosis and leptospirosis cases, considering Portuguese edaphoclimatic conditions in early autumn [34].

A wide range (1–20 days) of hospitalization length was detected in our study and in fact, usually, longer hospitalizations are needed in critical or septic cases, and shorter hospitalizations are needed in mild affected patients [8,12,28].

For parvovirosis, young age and lack of vaccination have been identified as increasing risk factors [8,24]. Many studies also detect increased risk in selected breeds as well as in intact males [8,24,30]. The presence of concomitant infections such as gastrointestinal parasites might be another risk factor for the increased severity of parvoviral enteritis [8,24].

In our study, the median age was 4 months old and young age (<2 years old), and this age group was significantly associated with an increased risk of being hospitalized at the BICU with parvovirosis. These results are consistent with the known epidemiological factors of parvovirosis which mainly affects puppies aged between 6 weeks and 6 months [8,24,30,35,36]. Some previous studies also found more cases and increased mortality among male patients [36,37], especially in intact males older than 6 months [8,24]. CPV can affect any susceptible dog, but different studies point to some breeds with increasing disease risk [8,24,30]. In our study, the presence of more than 50% of pure breeds suggests more susceptibility to the disease, but again this suggestion can be biased by the owner’s capacity of supporting the underlying veterinary care.

We found only one fully vaccinated dog hospitalized with parvovirosis. Incomplete vaccination was significantly associated to an increased risk of being hospitalized at the BICU with parvovirosis. Vaccination against CPV-2 is a core component in a dog’s vaccination schedule [17]. There is a high correlation between complete vaccination and the development of a robust and long-lasting immunity against CPV [8,9,17]. These results reinforce the importance of a complete vaccination against this virus. Most of the CPV cases in our BICU had no concomitant disorders. This finding contrast with a study in which the presence of gastrointestinal parasites seems to increase the severity of parvovirosis [8,36]. Moreover, parvovirosis alone is sufficient for hospitalization and eventual death [8].

The prevalence of parvovirosis typically increases during warm months [24]. In line with this previous evidence, we found an increase in cases in September and October, corresponding to the end of summer and autumn months, usually warm and dry in Portugal [34]. In our study, warm months registered most of parvovirosis cases although not significantly in the multiple analysis (Table 3). Interestingly, there were registered cases in all months. This might be explained by a temperate climate of the Lisbon area, which records an average temperature of 8 °C–15 °C in winter and 18 °C–28 °C in the summer [34]. A peak incidence of parvovirosis during summer months described in some studies is associated with an increasing rate of outdoor activities by dogs, increasing the chance of viral exposure [8,24,35,36,37].

Usually mild to severely affected dogs require hospitalization [30,38]. In our study, the median hospitalization period was 4.0 days, and this indicator goes up to 5.0 days considering only discharged patients. Similar results were obtained by Kalli et al. (2010). Lethal cases by parvovirosis can occur just 2 days after the manifestation of clinical signs, usually associated with sepsis [30]. Generally, infected dogs who survive the first 3–4 days of hospitalization are able to fully recover [24]. This explains the increment on the median hospitalization period when cases by death and euthanasia were excluded.

The survival rates of canine parvovirosis range from 9% in untreated animals to 80–90% with aggressive treatments in specialized referral hospitals [24,38]. Savigny and Macintire (2010) mention survival rates above 75% in specialized care facilities and even ranging 96–100% with 24 h care and access to therapeutic resources, such as blood transfusion, compared with nonspecialized care veterinary clinics, with survival rates varying from 67 to 75% [39]. The survival rate for parvovirosis cases in our study was 84%, consistent with results from specialized referral hospitals.

Leptospirosis has a heterogenous clinical presentation and outcome in dogs. The most frequent disease manifestation among hospitalized patients is a subacute infection with renal together with hepatic failure [10]. There is inconclusive evidence concerning age as a potential risk factor [40,41]. Intact males, large breeds, and mixed breeds particularly from working dog populations might be at a higher risk of the infection due to environmental exposure [42]. Given the age, vaccination, breed, sex, and neuter are referred widely among the literature [10,41,42,43]; all variables with *p* < 0.2 in individual analysis were kept in the final logistic regression model.

In our study, the median age was 6.0 years old and being from the adult age group (≥2 and <10 years old) was significantly associated with an increased risk of being hospitalized at the BICU under this condition. The median age and the modal age group were similar to previous studies [41,44,45]. Like other studies [46,47], we found a similar sex proportion and an over-representation of intact dogs without statistical significance. There is consistent evidence of a higher risk for leptospirosis in intact males, namely working dogs in published studies [40,41,43]. This evidence suggests an eventual disease association with hormone levels and risky behaviours such as smelling urine [41,45]. Pure-breed dogs were predominant in our data but with no statistically significant differences. This is unexpected given that it is known that leptospirosis can affect dogs of any breed [10,42]. However, large breeds and mixed breeds seem to be at a higher risk of exposure and disease [42,43]. A study identified a higher risk of hospitalization in small breed dogs, suggesting a closer owner–animal relationship as a reason for seeking medical care and supporting hospitalization [40].

We found a higher proportion, although without statistical significance, of leptospirosis when the vaccination status was not updated. Two reasons are suggested for the lack significant results for the protective effect of vaccination in areas with a high risk of exposure, such as Portugal [17,18]. Leptospirosis vaccines provide a shorter duration of immunity than core vaccines requiring annual boosters [17]. Therefore, a delay in vaccination may increase the chance of exposure for the dog to the disease. In addition, leptospirosis was reported in dogs immunized with bivalent vaccines, unable to provide cross-protection against other serovars circulating in Europe [40]. In our work, most leptospirosis cases had no concomitant disorders, and its presence was significantly identified as potentially reducing the risk of hospitalization due to leptospirosis. We suggest that the severity of acute leptospirosis cases alone justifies hospitalization, contrary to controls where concomitant disorders are frequent, and then it might bias the analysis. Data on concomitant disorders are scarce in studies concerning canine leptospirosis [40,41].

Usually, the case incidence increases with rainfall and warm conditions [40,43]. Our results suggest a seasonal trend in the incidence of leptospirosis with an increased number of cases during autumn, winter, and spring, as March lists the record of admissions followed by a drop during summer with no cases in August. Rainfall increases in Portugal from October to May, associated with mild temperatures, contrary to typical warm and dry summers from June to September [34]. Although, in our study no statistically significant differences comparing warm and cold months were identified in the multiple analysis (Table 3). Further studies concerning rainfall period, even comparing different years, could be performed as other studies also link higher rainfall to an increased incidence [25,40,43]. It is considered that a seasonal ID related to warm conditions and rainfall increases favour the survival of leptospires in stagnant waters [41,48], not affecting the viability of rodents, the most important reservoir, highly adaptable to changing environments [49].

In our study, the median hospitalization period of leptospirosis was 5 days (all patients) and 7 days (discharged patients), lower than 8 and 10 days reported elsewhere (Hartskeerl et al., 2011). Contrary to Portugal, the Netherlands is a country in which the notification of canine leptospirosis cases is mandatory [25]. Therefore, we can speculate that the obligation of notification encourages the choice to support longer hospitalization periods. The survival rate at discharge was 54.4%, within the survival rates from 52% to 68% reported by other studies performed in Europe [41,46]. In facilities with accessible haemodialysis, the reported survival rates can go higher than 80% [10,50]. Haemodialysis is not yet available at BICUs, which may explain the differences in survival rates reported by these facilities.

Current evidence is inconclusive whether age, sex, neuter status, and breed are indeed risk factors for MDR pathogens [51,52] as there is a lack of statistical evidence, probably due to the diversity of clinical presentations of MDR infections (Table A2) [52,53]. In both human and veterinary medicine studies, the presence of chronic diseases, chemotherapy treatments, previous antimicrobial treatments, surgery, and long hospitalization periods are recognized as predisposing factors to MDR infections [51,52,54,55].

A median age of 9.5 years among MDR-infected patients represents the highest among the studied diseases, with 50% of cases occurring in the older age group (≥10 years old). Being in this age category was significantly associated with an increased risk of hospitalization at the BICU with an MDR infection. Gibson et al. (2008) found similar median ages and ranges and Tenney et al. (2018) identified old age as a risk factor for urinary tract infections caused by MDR; but in other study, MDR urinary tract infection (UTI) was not significantly associated with age [51]. Human medicine studies identify geriatric patients as the most affected by MDR infections [54]. Regardless of higher proportions of males, intact, and breed dogs, none of these variables was statistically significant in our models. Additionally, Qekwana et al. (2018) found a higher frequency without statistical association of male dogs presenting MDR UTIs. La Fauci and Alessi (2018) also identified a higher frequency of MDR infections in human male patients, but a previous study described being female as a risk factor for UTI in general [51,53]. No breed association was identified in previous studies concerning MDR infections in pets [51,52].

There was a higher proportion of vaccinated dogs, with the recommended vaccines referred in the guidelines [17] among MDR-infected patients compared to the other diseases investigated in our study (Table 2). Two reasons that explain this proportion are the older age of the cases and the number of chronic cases with previously diagnosed concomitant diseases suggesting a population with regular medical care. Not updated vaccine status was identified as potentially reducing the risk of hospitalization due to MDR infection. Information on the vaccination status of MDR-infected patients is scarce in studies [51,52,55], but it is referred to a lower number of MDR cases among non-vet visiting dogs [55] and a higher risk of MDR in cases of frequent veterinary visits [56]. These results should be an alert to reinforce the prevention of nosocomial bacterial infections through strict disinfection protocols, and, for example, having designed areas or consultation rooms exclusively for routine/prophylactic procedures strictly disinfected and physically separate from other hospital areas.

Most of the MDR cases in our study had a concomitant disorder. The presence of a concomitant disorder was significantly associated with an increased risk of being hospitalized at the BICU with MDR infection. According to previous studies, indicators of the presence of concomitant disorders as prolonged hospitalizations, concurrent chemotherapy, and previous antimicrobial therapy increase patients’ susceptibility for MDR infection [52,53,54].

Even though the population analyzed was small for seasonal trend identification (not statistically significative), a slight increase in admissions of MDR-infected patients was recorded in March but with cases in all months. Temperate climate spring conditions [34] in this month may help the growth and proliferation of mesophilic bacteria [57].

The median length of hospitalization was 3.0 days, the lowest duration among the investigated diseases with a range (1–13 days) not influenced by the inclusion of all animals or only the discharged ones. Gibson et al. (2008) obtained a median hospital stay of 10 days but considered hospitalization days pre-isolation, contrary to our study. MDR pathogens can cause infections in different animal tissues. They are difficult to clear with standard drugs, and their infections are usually associated with poorer prognosis [11]. In our study, the survival rate of 71.4% at discharge should be interpreted carefully, as mentioned, as many dogs had concomitant diseases that might influence the outcome, and the treatment of the primary condition can help in the resolution of the MDR infection [52]. The survival considering all admissions decreases to 53.6% due to the presence of chronic conditions.

Canine distemper virus (CDV) affects the gastrointestinal, respiratory, and neurologic systems and it can resurge in canine outbreaks with high mortality rates [12]. Young and non-vaccinated animals are typically at a higher risk [31,58]. Immunosuppression induced by CDV infection may favour concomitant disorders [59]. Putative risk factors, such as breed, sex, and neuter remain under debate in the literature [60,61,62]. Therefore, in our final logistic regression model, all variables with *p* < 0.2 in individual analysis were preserved. This features alongside models regarding CDV environmental dynamics [63].

Concerning distemper cases in our study, the median age was 1 year, and young age (<2 years old) was significantly associated with an increased risk of being hospitalized at the BICU with distemper. This is not a surprising result given that the incidence of CDV has been reported to be higher in young dogs [31,64]. Our population had an increased frequency of male and intact dogs but only being neutered was identified as potentially reducing the risk of hospitalization due to distemper. No protective effect of sex on CDV cases is consistent with the literature [61,62]. Yet, one study identified a higher prevalence in males and the authors discuss an association with steroids level [58], which can be lower in neutered animals. Distemper was the only disease investigated in our study with a higher frequency of mixed breeds than pure breeds, even though without achieving statistical significance. Most of our cases were infected during an outbreak with a likely origin in stray dogs captured and brought into Lisbon Metropolitan Region shelters [27]. This might explain the higher frequency in mixed breeds. In fact, some studies refer to a higher risk of infection in mixed breeds potentially due to less attention and care from their owners [58,62].

A not-updated vaccination status was significantly associated with an increased risk of being hospitalized at the BICU with distemper. This was an expected result given that vaccination induces good protection levels in fully vaccinated animals [17]. The proportion of vaccinated dogs in a population is very important for individual protection because herd immunity prevents the occurrence of future outbreaks [17,31]. Concomitant disorders were absent from 57.7% of the cases, a result not statistically significant in the final regression analysis. Some studies refer to the presence of concomitant disorders, namely respiratory bacterial infections, but do not mention them as risk factors for distemper [59,65,66,67].

As CDV is sensitive to high temperatures and dry conditions [31], most cases are expected to occur in cold and humid months [12]. In our study, the peak number of distemper cases was observed in January and December, the coldest and the most humid months in Portugal, and any case was registered in August or September, the hottest and driest months [34]; therefore, this monthly distribution of cases was expected in Portugal, even though without statistical significance.

The outcome depending on the clinical presentation varies from a complete recovery after an acute episode to a persistent disease with a reduced survival rate [31,68]. The survival rate at discharge (50.0%), and the median hospitalization period (3.5 days) are influenced by the nature of the clinical presentation such as the development of progressive neurological signs that greatly reduce the survival rate [12,31,69]. The development of new clinical signs weeks to months after a distemper clinical episode is frequent [31], which explains the substantial decrease in the final survival rate (34.6%) when considering readmissions. Based on our experience and according to other studies, it appears that, regardless of the clinical intervention, the disease progresses in a high proportion of dogs and animals that end up not surviving [12]. 

Based on our models, protocol recommendations are for early segregation of sick young dogs with incomplete vaccination history, and being attended to immediately at the isolation ward and hospitalized there, even before diagnostic test results. In addition, old dogs with recurrent hospital visits and concomitant disorders must be identified as potentially being MDR infected, be tested early with bacterial cultures and sensitivity tests, and sanitary measures must be reinforced with these patients.

Overall, some disease determinant factors were associated with an increased or decreased risk of infection and hospitalization of dogs and can be included in basic triage tools, to help clinicians and decision makers to improve vet care, efficiency in diagnosis, and treatment of infectious patients. As demonstrated in our study, the inclusion of age, vaccine status, and the presence of concomitant disorders in any dog infectious disease triage tool is fundamental. Despite requiring some improvement and more data, the inclusion of season and neuter status can be pertinent. The sex and breed were the variables with poorer performances and thus are less relevant considering the aims.

## 5. Conclusions

The final logistic regression models for parvovirosis, MDR infections, and canine distemper had a good predictive performance and, therefore, can be helpful in the early detection and triage of these patients. The model for leptospirosis was good although it can be improved for better performance as a triage tool. In particular, for highly infectious agents such as CDV and CPV, the use of these models can also contribute to prevent outbreaks. This study had some limitations due to its descriptive nature. First, the quality of the data collected is dependent upon several factors, such as the accuracy of the information shared by animal owners and the precision of the anamnesis registered by a variety of veterinary surgeons among the 7 years covered by the study. Second, reaching a definitive diagnosis depends on the clinical condition of the patient and the willingness of the dog owners to cover the respective costs. In addition, the economic situation of the households also affects the length of the hospitalization stay and potentially the outcome.

Overall, our study highlights that infectious diseases are still frequent in the canine population. This is a call for more effective preventive measures such as vaccination. Furthermore, knowing epidemiological factors associated with the disease can be a useful tool for early triage, screening, and diagnosis, crucial to the prediction and prevention of disease occurrence.

## Figures and Tables

**Figure 1 vetsci-10-00186-f001:**
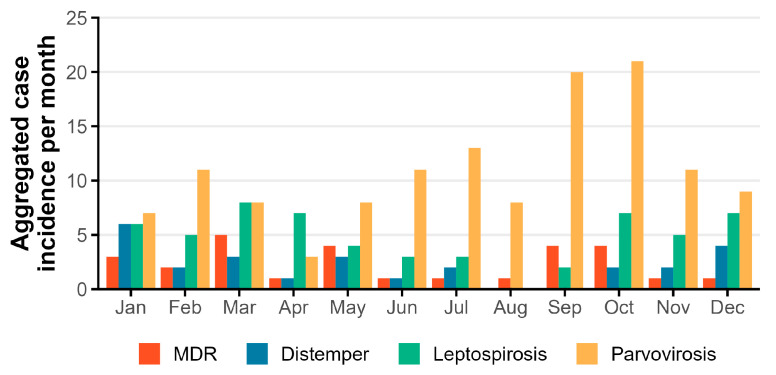
Aggregated case incidence per month for each main ID (parvovirosis, *n* = 130; leptospirosis, *n* = 57; MDR, *n* = 28; distemper, *n* = 26).

**Figure 2 vetsci-10-00186-f002:**
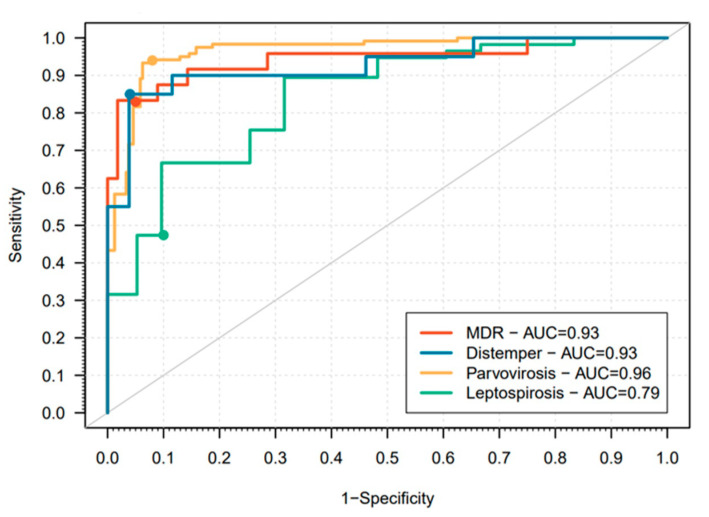
Receiver operating characteristic (ROC) curves for the final logistic regression models associated with each ID. The dots in the ROC curve are the points that led to the sensitivity and specificity estimates reported in the main text.

**Table 1 vetsci-10-00186-t001:** Diagnostic tests performed for the main ID.

Disease	Tests
Parvovirosis (*n* = 130)	Rapid immunomigration test (*n* = 62) ^1^ Real-time PCR (RT-qPCR) (*n* = 68)
Leptospirosis (*n* = 57)	Single dosage IgM measure by semi-quantitative indirect immunofluorescence for IgM detection (*n* = 54) ^2^ PCR on Blood/Urine (*n* = 3)
MDR (*n* = 28)	Bacterial culture and antibiotic susceptibility testing from the adequate tissue (urine, skin, effusion liquid, infected wounds, and others, *n* = 28)
Distemper (*n* = 26)	Serology for antibody IgM/IgG detection (*n* = 19) ^3^ Real time PCR (RT-qPCR) from oronasal, rectal, cerebrospinal fluid or blood sample (*n* = 6) Rapid immunomigration test (*n* = 1) ^4^

^1^ WITNESS^®^ Parvo rapid test specificity 99.5%; sensitivity 96%; ^2^ The cut-off for IgM detection is >1:200, test performed at DNATech^®^ laboratory; ^3^ The cut-off for IgM/IgG detection is >1:25, test performed at DNATech^®^ laboratory; ^4^ Quicking^®^ pet rapid test CDV Ag rapid test specificity 97.5%; sensitivity 98%.

**Table 2 vetsci-10-00186-t002:** Characterization of total infectious population and top 4 IDs by age, sex, neuter status, breed, vaccination status, concomitant diseases, hospitalization length, hospitalization outcome, and final outcome of the case.

Parameter	Category	All Infectious (*n* = 263)	Parvovirosis (*n* = 130)	Controls (*n* = 260)	Leptospirosis (*n* = 57)	Controls (*n* = 114)	MDR (*n* = 28)	Controls (*n* = 56)	Distemper (*n* = 26)	Controls (*n* = 52)
Median Age [range] Q1; Q3		0.8 [0.1–18] 0.3; 6.0	0.3 [0.1–14] 0.2; 0.48	8.0 [0.1–17] 4.0; 11.0	6 [0.2–15] 3.0; 8.0	6 [0.1–15] 2.0; 10.75	9.5 [0.4–16] 7.0; 12.0	0.6 [0.1–14] 0.2; 2.5	1 [0.2–15] 0.4; 4.75	8 [0.6–15] 5.0; 11.0
Age group (years) *n*(%)	<2	156 (59.3)	123 (94.6)	33 (12.7)	9 (15.8)	27 (23.7)	2 (7.1)	41 (73.2)	15 (57.7)	6 (11.5)
	≥2 and <10	81 (30.8)	6 (4.6)	137 (52.7)	45 (78.9)	48 (42.1)	12 (42.9)	10 (17.9)	9 (34.6)	28 (53.8)
	≥10	26 (9.9)	1 (0.8)	90 (34.6)	3 (5.3)	39 (43.2)	14 (50.0)	5 (8.9)	2 (7.7)	18 (34.6)
Sex n(%)	Female	112 (42.6)	56 (43.1)	120 (46.2)	28 (49.1)	55 (48.2)	9 (32.1)	26 (46.4)	8 (30.8)	25 (48.1)
	Male	151 (57.4)	74 (56.9)	140 (53.8)	29 (50.9)	59 (51.8)	19 (67.9)	30 (53.6)	18 (69.2)	27 (51.9)
Neuter Status *n*(%)	No	236 (89.7)	128 (98.5)	182 (70)	45 (78.9)	82 (71.9)	20 (71.4)	48 (85.7)	25 (96.2)	34 (65.4)
	Yes	27 (10.3)	2 (1.5)	78 (30)	12 (21.1)	32 (28.1)	8 (28.6)	8 (14.3)	1 (3.8)	18 (34.6)
Vaccination Status *n*(%)	Updated	39 (14.8)	1 (0.8)	78 (30.0)	15 (26.3)	29 (25.4)	12 (42.9)	6 (10.7)	3 (11.5)	17 (32.7)
	Not updated	195 (74.1)	119 (91.5)	162 (62.3)	39 (68.4)	79 (69.3)	12 (42.9)	50 (89.3)	17 (65.4)	35 (67.3)
	Unknown	29 (11.0)	10(7.7)	20 (7.7)	3 (5.3)	6 (5.3)	4 (14.3)	-	6 (23.1)	-
Breed *n*(%)	Breed	151 (57.4)	71(54.6)	162 (62.3)	35 (61.4)	72 (63.2)	21 (75.0)	32 (57.1)	12 (46.2)	36 (69.2)
	Mixed breed	112 (42.6)	59(45.4)	98 (37.7)	22 (38.6)	42 (36.8)	7 (25.0)	24 (42.9)	14 (53.8)	16 (30.8)
Concomitant Diseases n(%)	No	148 (56.3)	148(56.3)	77 (29.6)	36 (63.2)	32 (28.1)	3 (10.7)	32 (28.1)	15 (57.7)	11 (21.2)
	Yes	115 (43.7)	115(43.7)	183 (70.4)	21 (36.8)	82 (71.9)	25 (89.3)	82 (71.9)	11 (42.3)	41 (78.8)
Hospital stay length (days) Median [Range]	All	4.0 [1.0–20.0]	4.5 [1.0–18.0]	-	5.0 [1.0–16.0]	-	3.0 [1.0–13.0]	-	3.5 [1.0–20.0]	-
	Only Discharge	5.0 [1.0–20.0]	5.0 [1.0–18.0]	-	7.0 [1.0–16.0]	-	3.0 [1.0–13.0]	-	4.0 [1.0–20.0]	-
Hospital stay Outcome *n*(%)	Discharge	189 (71.9)	109 (83.8)	-	31 (54.4)		20 (71.4)	-	13 (50.0)	-
	Euthanasia	40 (15.2)	1 (0.8)	-	18 (31.6)	-	7 (25.0)	-	11 (42.3)	-
	Dead	34 (12.9)	20 (15.4)		8 (14.0)		1 (3.6)		2 (7.7)	
Final Outcome *n*(%)	Discharge	181 (68.8)	107 (82.3)	-	29 (50.9)	-	15 (53.6)	-	9 (34.6)	-
	Euthanasia	46 (17.5)	1 (0.8)	-	18 (31.6)	-	12 (42.9)	-	15 (57.7)	-
	Dead	36 (13.7)	22 (16.9)	-	10 (31.6)		1 (3.6)		2 (7.7)	-

**Table 3 vetsci-10-00186-t003:** Results of multiple logistic regression models for all 4 diseases. where the symbol “-“ indicates the reference category used in each covariate.

Disease	Parvovirosis	Leptospirosis	MDR	Distemper
Variable	*p*-value Estimate Std error	*p*-value Estimate Std error	*p*-value Estimate Std error	*p*-value Estimate Std error
Age group				
<2 years	<0.001 4.26 0.53	-	-	<0.001 3.78 1.17
≥2 to and <10 years	-	<0.001 1.65 0.50	0.16 1.59 1.14	-
≥10 years	0.43 −0.89 1.13	0.6 −0.40 0.77	<0.01 3.51 1.19	0.62 −0.62 1.26
Sex				
Female	-	-	-	-
Male	-	-	-	0.6 −0.43 0.86
Neuter Status				
No	-	-	-	-
Yes	0.09 −1.65 0.98	-	0.56 −0.53 0.92	0.04 −3.25 1.55
Vaccination Status Updated				
Yes	-	-	-	-
No	<0.001 4.25 1.09	-	0.02 −2.07 0.85	0.03 2.70 1.26
Breed				
Breed	-	-	-	-
Mixed breed	0.98 −0.01 0.43	-	-	0.11 1.27 0.8
Concomitant Disorders				
No	-	-	-	-
Yes	0.06 −0.82 0.43	<0.001 −1.62 0.41	0.03 2.39 1.10	0.3 −0.89 0.84
Season				
Cold (November-April)	-	-	-	-
Warm (May-October)	0.5 0.25 0.43	0.07 −0.68 0.38	-	-

## Data Availability

The datasets used and/or analyzed during the current study are available from the corresponding author upon reasonable request.

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
