# Peer review of "Epidemiologic Factors Supporting Triage of Infected Dog Patients Admitted to a Veterinary Hospital Biological Isolation and Containment Unit"

_vetsci, 2023, doi:10.3390/vetsci10030186_

Round 1

Reviewer 1 Report

This article covers a really interesting topic, and could be a valuable contribution to veterinary hospital infectious disease risk minimisation as an exemplar use of predictive modelling in risk mitigation strategies. However, I don't think that is conveyed sufficiently in the manuscript yet, and some of the statistical methodology required modification to ensure validity.

Major comments

-       Study objectives need to stated clearly upfront in the introduction. The current stated aims are not specific enough to stand alone.

-       As this is in the format of a case-control study, review of the STROBE guidelines, and more formalised reporting in accordance with these guidelines, is recommended.

-       Overdispersion is a considerable influence on the findings of at least three of the regression models (parvovirus, MDR, distemper) - this should be remedied before publication, with reporting of model diagnostics that indicate so.

-       The discussion is far too long and dissociated from the stated aims of the research. The stated aims of the study specifically relate to the triage of patients in the BICU and development of a patient management tool; yet the discussion is almost entirely focussed on making wide-ranging causal inferences that are not appropriate given the study population and the statistical approach undertaken (predictive modelling, rather than the required  explanatory approach for causal inferences). While the Discussion opens with a statement that the study 'aimed to characterise the general infectious population admitted at a University -based BICU', this aim is not specified in the more general aims at the start of the study. Indeed, such study findings are unlikely to be of interest to a broad population of readers, as the results at BICU are not generalisable to any other hospital- it is simply background information for context and consideration against the model results. The aspects of the article that are more likely to be of broad interest relate to demonstration of the use of such data for predictions that inform patient management tools and triage in a hospital setting to minimise the risk of disease spread (as per the study aims referenced in the introduction).

- In line with the above comment, I expected to see results relating to application of these predictive models to protocols at BICU, or discussion focusing on how these predictions are intended to be applied- that is where the real interest of this work lies.

Minor comments

-       Introduction: In considering the SOP in the introduction, it is not made clear that cleaning and disinfection protocols fall under the essential criteria.

-       Introduction: I would recommend making it clear that the discussion on controlling spread of infections relates to prioritised diseases. For example, irrespective of the immediate clinical problem at hand, all patients presenting are likely to be infected with least one type of parasite, irrespective of its contribution to the current state of disease. Hence, segregation of ‘infected patients’ in the absence of specification of disease(s) technically results in an assumption that all patients are put in quarantine.

-       Introduction: in what locations and what type of hospitals have parvovirus et al. been reported as common IDs? This should be clarified.

-       Introduction: reference is made to a “reasonable amount’ of guidelines—it would be better to be specific. What location and types of animal care facilities fall under the scope of having a ‘reasonable amount’ of guidelines, and is the ‘reasonable amount’ related to the scope of guidelines, or the coverage of the relevant population with these guidelines?

-       Introduction: remove the results summarised in the last few lines.

-       Methods: Section 2.1: If dogs admitted to the BICU are routinely screened for ID, specify the details here.

-       Methods: it would be helpful to have an idea of how an ‘inconclusive’ result is defined- is that inconclusive as reported by a lab, or as defined on instructions for a point-of-care test? Or does it involve clinical interpretation pertaining to test sensitivity, i.e. a negative result from a clinically suspicious dog? This information could be included in Table 1, alongside any published data regarding test sensitivity and specificity.

-       Methods: describe ‘definitive ID diagnosis’- does this involve clinical diagnosis, diagnostic testing, or both? If clinical features were part of the characterisation of a ‘definitive diagnosis’, describe what they were for each disease.

-       Methods: specify the approach to random selection of controls from the BICU population of dogs. Was eligibility for controls dependent on not having an infectious disease diagnosis, and if so were they screened for disease using the same tests as the cases before being confirmed as controls? Was selection of controls specific for each major disease (e.g. were dogs infected with leptospirosis but not parvovirus eligible for inclusion as controls for the parvovirus?) etc.

-       Methods/ Table 2: interquartile ranges should be presented as an indicator of data skewness 

-       Results: lines 165 to 175, I suggest stating upfront that these results pertain to the infected disease dogs, to ensure clarity.

-       Table 2 comprehensively covers the descriptives for the study; so I’d suggest removing all the repetition of these results spread throughout the text in the Results section (including the analytics section), for brevity.

-       Table 2 indicates some considerable differences between the disease groups and their respective controls (e.g. parvovirus – median age of cases 0.3; median age of controls 8.0). The potential influence of this on the respective models should be specifically discussed.

-       Table 3 could be simplified- perhaps by putting the univariable results in supplementary material and leaving the multivariable results in the main text

-       In table 3, I’d suggest putting 1 as the baseline value (rather than ‘-‘), to aide readers’ interpretation—particularly those that are not familiar with interpretation of regression models

- Figure one is descriptive, so I’d suggest moving it to the start (descriptive) part of the results. 

Reviewer 2 Report

The paper is a scientific work reporting the most frequent infectious diseases recorded during seven years at the BICU in Lisbon. The paper is easy to read and to understand.

Brief informations about the methods used for the confirmation of ID should be reported:  Authors should report both the manufacturer and cut off for the IgM leptospirosis IFAT, manufacturer of rapid CPV immunomigration test, both the manufacturer and cut off for CDV serology.

Authors should explain in which way was confirmed the diagnosis of leptospirosis: a single dosage of IgM or a time course?

Leptospirosis: Authors reported that vaccination status was not significantly associated with the disease (lines 221-222). But at lines 373-377 they report a higher proportion of leptospirosis when vaccination status was not updated. These sentences seems contradictory. Authors believe that a more frequent booster may improve protection (e.g.  6 months)?

Tables 2 and 3 should be better formatted

Reviewer 3 Report

The manuscript is enough well written, clear and has a good structure. 

The topic is very interesting, anyway some changes have to be performed  before going ahead in the pubblication phase. If authors well follow the suggestion given below I will certainly recommend this case report for pubblication on the Journal.

At the beginning, I advice you to report some articles about similar study in the introduction to give the reader a global view of the topic. For this reason, You can include these works:

https://doi.org/10.2460/javma.245.8.923

https://doi.org/10.1111/vec.12885

Then, I aslo suggest you to include this recent work about distemper virus in foxes that propose an integrated tentative remote sensing approach based on NDVI entropy to model the disease: 

- https://doi.org/10.3390/ani12081049

Reviewer 4 Report

The paper entitled -  Epidemiologic factors supporting triage of infected dog patients admitted to a Veterinary Hospital Biological Isolation and Containment Unit’ presents some epidemiological data obtained during 7-year period.

The idea is interesting but the whole paper is not clear and difficult to trace.

Simple summary, abstract (and actually material and methods)

Line 27, Line 40, Specify what means MDR – provide the additional, more detailed information about bacterial infections, even as supplementary materials.

Line 31 - “Our results, by improving the knowledge about epidemiology and clinical presentation of these diseases, show the value of collection, analysis and share of clinical data and contribute for the creation of infectious cases triage tools as algorithms” – How your results improving knowledge ? We know that non-vaccinated dogs are more prone to diseases like CPV-2, and older dogs suffer more frequent from bacterial infection. You mentioned about algorithm, give more details.

Lines 46 – 49  - Truism. How we can use your model in vet practice ? Explain it more extensively in the final part of discussion

Introduction – Generally, this part of the paper is quite good. Authors give sufficient background and justification of the research.

Authors should specify the goal of study more clearly. Did they want to provide tools to patient management ? Analyze the epidemiological factors ? Provide algorithm to triage of canine patients ?

 Line 90-93 – This part of text sounds rather like conclusion

Material and methods

Line 113 – Provide the full data concerning the tested dogs as supplementary materials. Their vaccination status (both whole control and infected group), and all data that you use in the paper

Line 126 - Why you did not decide to include one total negative group for all infectious diseases ?

Line 131 – Again, provide detailed data for each dog

Line 134 – Heading should be placed over the table.

Table 1 – Sum of tested samples is higher than ‘n’ for distemper - 32 vs 26.

Results

Generally present your data in some other way, because these tables are difficult to trace.

Prepare one table presenting tested groups and control, give median, range verify if differences between particular infected groups (age group, neuter status etc.) and control groups are significant.

If you prepare the regression analysis, clearly point the risk factors for each group with infection, provide OR, 95% CI and p-value and make the table more clear, because the presented one is chaotic.

Line 241 – This issue should be properly discussed in discussion section.

Maybe it is also worthy to include the equation with the significant coefficients, as conclusion for each model ? You should also include the season in your model, as it may also give the additional information about the prevalence of the infections.

Table 3 – There is a lot of gaps. Is it intentional ?

Discussion

This section is well written. It is slightly too long, but interesting to read. Authors should also use more latest papers.

Lines - 288-290 – This also should be also included in detailed characteristics of dogs. How many dogs were not fully vaccinated due to the age ? They should be also be analyzed as separate, additional group.

Lines - 380-381 – What do you mean as ‘protection factor’ ? The presence of leptospirosis may protect from the other disorders ?

Lines 430-431 – Explain it more extensively. Why lack of full vaccination status may be connected with the lower prevalence of MDR ?

Line 437  - Is this result included in the any of tables ?

Lines 443-444 – So, how you explain the drop in MDR cases in April and higher prevalence in May ?

Lines 474-476 – I think that this is not the main cause, as mix breeds dogs also can be subjected proper attention and care. Please, try to find additional argument.

Conclusion is quite fair, Authors pointed the limitations. This part is much better balanced than concluding part of the simple summary, which is a bit over the top.

Concluding, paper may be valuable for the vets. It gives some insight into the risk factors of particular disease. Maybe whole paper is not inventive, but it contains piece of information which may appear useful in practice. There are too much question marks to publish paper in the present form. Authors should provide extensive supplementary materials presenting detailed information about dogs.

Presentation of results must be also improved, as it is very difficult to trace results. There are also many other comments that I have included above. My decision about the manuscript is reconsideration of paper after major revision.

Round 2

Reviewer 1 Report

The two key concerns relating to this manuscript have not been resolved.

Firstly, the modelling issue has not been resolved. Communication issues as to the cause aside,
erratically wide confidence intervals are common in the results presented here (for example, 70.3(12.2 - 1349.8); 43.8 (6.40 – 930.9) and 70.7 (95% CI 27.2 – 224.3)). They represent considerable underlying problems in unbalanced study data/model stability, and so these figures are invalid as well as being uninterpretable (a 95% CI of 12.2 - 1349.8 provides no useful information as to what they true population value is likely to be). Recruit a statistician to resolve the issues. 

Secondly, in my opinion the manuscript has not improved in terms of capturing the interest of a broader set of readers. This has already been commented on in the previous review; to reemphasise here, it may be considered unlikely that a broad readership are particularly interested at the risk profile of dogs at this particular veterinary hospital that is unknown to them and their work. However, there may be broader interest in how taking a risk profile at a particular hospital can feed into development of an infectious disease control program that can improve patient management and biosecurity outcomes, as this can provide an exemplar approach that could be replicated elsewhere. In view of this, the discussion misrepresents where the value of these data lies, as well as remaining far too long and irrelevant. There is no discussion at all on how these data can be applied to inform the development of an infectious disease control program. It remains entirely based around observations that are not particularly interesting and/or can be considered to have exceptionally limited external validity given the study population. Most particularly, comments around the protective effects of vaccination are entirely inappropriate: "However, vaccination seems not to provide significant protection against this infection [leptospirosis]" (line 383) —it is entirely inappropriate to be drawing such inferences outside of the context of a vaccine effectiveness trial. 

Author Response

Dear Alan Wang,

Thank you for your letter and the opportunity to revise our paper entitled “Epidemiologic factors supporting triage of infected dog patients admitted to a Veterinary Hospital Biological Isolation and Containment Unit”.

The authors would like to thank the reviewers for the careful and through revision of this manuscript and for the constructive suggestions and considered our previous changes.

Therefore, we have carefully studied the new reviewers’ comments and revised the manuscript accordingly.

Please find below a point-by-point response to the reviewers’ comments.

Reviewer 1:

The two key concerns relating to this manuscript have not been resolved.

Firstly, the modelling issue has not been resolved. Communication issues as to the cause aside, erratically wide confidence intervals are common in the results presented here (for example, 70.3(12.2 - 1349.8); 43.8 (6.40 – 930.9) and 70.7 (95% CI 27.2 – 224.3)). They represent considerable underlying problems in unbalanced study data/model stability, and so these figures are invalid as well as being uninterpretable (a 95% CI of 12.2 - 1349.8 provides no useful information as to what they true population value is likely to be). Recruit a statistician to resolve the issues. 

  • Author’s response: We thank the reviewer for raising this point. We went back to our model outputs, and we found that the standard errors of the coefficients (i.e., log(ODs) are not erratic. As such, the confidence intervals for the log(ODs) are reasonable in width, but they become wider when applying the exponential transformation to the lower and upper bounds of these CIs, giving a false impression of an eventual model instability. To avoid this interpretation, we have now revised Table 3 to report only the model estimates, the respective standard errors/p-values for each estimated model.

Secondly, in my opinion the manuscript has not improved in terms of capturing the interest of a broader set of readers. This has already been commented on in the previous review; to reemphasise here, it may be considered unlikely that a broad readership are particularly interested at the risk profile of dogs at this particular veterinary hospital that is unknown to them and their work. However, there may be broader interest in how taking a risk profile at a particular hospital can feed into development of an infectious disease control program that can improve patient management and biosecurity outcomes, as this can provide an exemplar approach that could be replicated elsewhere. In view of this, the discussion misrepresents where the value of these data lies, as well as remaining far too long and irrelevant. There is no discussion at all on how these data can be applied to inform the development of an infectious disease control program. It remains entirely based around observations that are not particularly interesting and/or can be considered to have exceptionally limited external validity given the study population.

  • Author’s response: We thank the reviewer for this comment. We revised the discussion to clarify, based on the previous question, some of the increased and decreased risk factors for each disease. We think this way is clearer and more useful for readers. We also included a global discussion/contribution of our work for triage models: “Overall, some diseases determinant factors were associated to an increased or de-creased risk of infection and hospitalization of dogs and can be included in basic triage tools, to help clinicians and decision makers to improve vet care, efficiency in diagnosis and treatment of infectious patients. Demonstrated in our study it is fundamental the inclusion age, vaccine status and the presence of concomitant disorders in any dog infectious disease triage tool. Even though requiring some improve and more data the inclusion of season and neuter status can be pertinent. The sex and breed were the variables with poorer performances and so less relevant considering the aims.” (lines 621-629).

Most particularly, comments around the protective effects of vaccination are entirely inappropriate: "However, vaccination seems not to provide significant protection against this infection [leptospirosis]" (line 383) —it is entirely inappropriate to be drawing such inferences outside of the context of a vaccine effectiveness trial. 

  • Author’s response: We thank the reviewer for this remark. In fact, the authors agree that this sentence could be misinterpreted, so we removed this sentence and clarified the next: “Two reasons are suggested for the lack significant results  of the protective effect of vaccination in areas with a high risk of exposure, such as Portugal” (line 462-64)

We believe this is an improved version of the original manuscript and we look forward to hearing your decision soon. If you have any additional query, do not hesitate to contact us again.

Sincerely,

Solange Gil

DVM, Assistant Professor

Reviewer 4 Report

I would like to thank Authors for the revised version of the manuscript. They addressed to most of my comments and suggestions. Nevertheless, there are still some important issues that should  be explained more extensively.

 Firstly, I think that one of the target group of the paper are clinicians. Therefore your material (analyzed group) should be perfectly described. When I mentioned about MDR, I don’t require the explanation of definition (that you have included in the text). Please give more details about MDR infections (prevalence of particular genus of bacteria, aerobic/anaerobic, Gram staining) – such information are valuable in context of MDR infections and are welcomed as supplementary materials.

The second issue

  You wrote:

 ‘This study serves as a background point to create algorithms for triage. Algorithms and trained models are supported by consistent data’

 ‘To achieve that, the specific aims of this work were to characterize  the infectious dog population admitted to an University-based Biological Isolation and Containment Unit (BICU) over a 7-year period and to identify various determinant factors  for early detection and hospitalization of dogs with ID.’

I appreciate that the Authors edited the goal of the study. The present version better displays the content of the paper. However, you wrote about characterization of the population, you wrote about the data that can be included in the model. We need more details about dogs included in the study.

 You claim that data some part of data are confidential, I agree they can be skipped if they identify dogs or owners, but clinical data should be included. Please provide the supplementary table with detailed information about dogs (data such as gender, age, vaccination status, and additional clinically relevant data). Such information may be really precious when it comes to model or algorithm preparations.

The third issue, it is slightly connected with the previous. The detailed information of the tested group may give us better insight into the results. I have mentioned about negative group that you included in the study. Your data demand some additional glimpse. You obtained very broad confidence intervals in some parameters what makes the results difficult to interpretation and to transfer the findings into practice (or apply into the model). This issue should be also explained and included in the text.

Final issue – I understand your results point that not vaccinated dogs have a lower frequency of MDR infections. Here the crucial issue is proper interpretation. The statement put out of the context may be dangerous as it may suggest that if you don’t vaccine dog you reduce the risk of MRD infection. Moreover, it may interpret – want to reduce the infection risk ? Avoid vet visits. This is very subtle issue and should be pointed very clearly and to not leave opportunity to misinterpretation.

 Minor:

Check the format of the tables, try to better adjust it to the page.

Some double spacing – for example  line 204, 247, 267, 297

Formatting 312-313.

Double dot 252, 312

 I have presented the doubts that should be explained. Nevertheless, I would like to appreciate some corrections made by the Authors.

-           The aim of the study is presented far more clear and is better adjusted to the paper.

-          Details (sensitivity and specificity) about diagnostics kits that you have included are welcomed.

-          I appreciate that you included season as factor in analysis.

-          Recommendations – lines 513-517 are really good idea (even if especially the first one seems to be obvious) it may be even more extended to make the paper more applicable.

Concluding, some doubts were explained, some issues were improved. Unfortunately, there are still some question marks, that should be discussed.

Author Response

Dear Alan Wang,

Thank you for your letter and the opportunity to revise our paper entitled “Epidemiologic factors supporting triage of infected dog patients admitted to a Veterinary Hospital Biological Isolation and Containment Unit”.

The authors would like to thank the reviewers for the careful and through revision of this manuscript and for the constructive suggestions.

We are pleased to acknowledge that the reviewers considered that the work described herein this paper “really interesting topic, and could be a valuable contribution to veterinary hospital infectious disease risk minimisation”.

Therefore, we have carefully studied reviewers’ comments and revised the manuscript accordingly.

Please find below a point-by-point response to the reviewers’ comments.

Reviewer 4

I would like to thank Authors for the revised version of the manuscript. They addressed to most of my comments and suggestions. Nevertheless, there are still some important issues that should  be explained more extensively.

 Firstly, I think that one of the target group of the paper are clinicians. Therefore your material (analyzed group) should be perfectly described. When I mentioned about MDR, I don’t require the explanation of definition (that you have included in the text). Please give more details about MDR infections (prevalence of particular genus of bacteria, aerobic/anaerobic, Gram staining) – such information are valuable in context of MDR infections and are welcomed as supplementary materials.

  • Author’s response: We thank the reviewer for this suggestion. The authors included in Appendix A the table A2 with data concerning MDR infected cases, organized by agent and type of infection, and mention to that in results section: lines 322-23 “Data concerning types of infection and bacterial organisms is presented in table A2.”

The second issue

  You wrote:

 ‘This study serves as a background point to create algorithms for triage. Algorithms and trained models are supported by consistent data’

 ‘To achieve that, the specific aims of this work were to characterize  the infectious dog population admitted to an University-based Biological Isolation and Containment Unit (BICU) over a 7-year period and to identify various determinant factors  for early detection and hospitalization of dogs with ID.’

I appreciate that the Authors edited the goal of the study. The present version better displays the content of the paper. However, you wrote about characterization of the population, you wrote about the data that can be included in the model. We need more details about dogs included in the study.

 You claim that data some part of data are confidential, I agree they can be skipped if they identify dogs or owners, but clinical data should be included. Please provide the supplementary table with detailed information about dogs (data such as gender, age, vaccination status, and additional clinically relevant data). Such information may be really precious when it comes to model or algorithm preparations.

  • Author’s response: We thank the reviewer for this suggestion. As suggested, the authors included a table with data concerning studied population in supplementary materials (table S1), and have mentioned this fact in the manuscript: “6. Patents Supplementary Materials: The following supporting information can be downloaded at: www.mdpi.com/xxx/s1, Table S1: Epidemiological data from the studied population” (lines 646-648)

The third issue, it is slightly connected with the previous. The detailed information of the tested group may give us better insight into the results. I have mentioned about negative group that you included in the study. Your data demand some additional glimpse. You obtained very broad confidence intervals in some parameters what makes the results difficult to interpretation and to transfer the findings into practice (or apply into the model). This issue should be also explained and included in the text.

  • Author’s response: We thank the reviewer for raising this point. We went back to our model outputs, and we found that the standard errors of the coefficients (i.e., log(ODs)) are not erratic. As such, the confidence intervals for the log(ODs) are reasonable in width, but they become wider when applying the exponential transformation to the lower and upper bounds of these CIs, giving a false impression of an eventual model instability. To avoid this interpretation, we have now revised Table 3 to report only the model estimates, the respective standard errors/p-values for each estimated model.

Final issue – I understand your results point that not vaccinated dogs have a lower frequency of MDR infections. Here the crucial issue is proper interpretation. The statement put out of the context may be dangerous as it may suggest that if you don’t vaccine dog you reduce the risk of MRD infection. Moreover, it may interpret – want to reduce the infection risk ? Avoid vet visits. This is very subtle issue and should be pointed very clearly and to not leave opportunity to misinterpretation.

  • Author’s response: We thank the reviewer for this remark. In fact, the authors agree that this sentence could be misinterpreted, so we removed the sentence “In addition, vaccinated animals tend to visit veterinary facilities more frequently.” And added: “These results should be an alert to reinforce prevention of nosocomial bacterial infections through strict disinfection protocols, and, for example having designed areas or consultation rooms exclusively for routine/prophylactic procedures strictly disinfected and physically separate from other hospital areas.” (lines 530-534)

 Minor:

Check the format of the tables, try to better adjust it to the page.

  • Author’s response: We thank the reviewer for this remark, we proceeded as requested

Some double spacing – for example  line 204, 247, 267, 297

Formatting 312-313.

Double dot 252, 312

  • Author’s response: We thank the reviewer for this remark, we proceeded as requested

 I have presented the doubts that should be explained. Nevertheless, I would like to appreciate some corrections made by the Authors.

-           The aim of the study is presented far more clear and is better adjusted to the paper.

-          Details (sensitivity and specificity) about diagnostics kits that you have included are welcomed.

-          I appreciate that you included season as factor in analysis.

-          Recommendations – lines 513-517 are really good idea (even if especially the first one seems to be obvious) it may be even more extended to make the paper more applicable.

  • Author’s response: We thank the reviewer for this positive comments. We proceeded to an extension to these recommendations as well: “Overall, some diseases determinant factors were associated to an increased risk of infection and hospitalization of dogs and can be included in basic triage tools, to help clinicians and decision makers to improve vet care, efficiency in diagnosis and treatment of infectious patients. Demonstrated in our study it is fundamental the inclusion age, vaccine status and the presence of concomitant disorders in any dog infectious disease triage tool. Even though requiring some improve and more data the inclusion of season and neuter status can be pertinent. The sex and breed were the variables with poorer performances and so less relevant considering the aims.” (lines 612-619).

We believe this is an improved version of the original manuscript and we look forward to hearing your decision soon. If you have any additional query, do not hesitate to contact us again.

Sincerely,

Solange Gil

DVM, Assistant Professor